# Clinical characteristics of paediatric autoimmune hepatitis at a referral hospital in Sub Saharan Africa

**Taiba Jibril Afaa** [1]*, **Kokou Hefoume Amegan-Aho**[2], **Matilda Tierenye Dono**[3], **Eric Odei**[4], **Yaw Asante Awuku**[5]

1 Department of Child Health, Medical School, College of Health Sciences, University of Ghana, Accra, Ghana, 2 Department of Paediatrics and Child Health, University of Health and Allied Sciences, Ho, Ghana, 3 Department of Child Health, Korle Bu Teaching Hospital, Accra, Ghana, 4 Department of Public Health, Korle Bu Teaching Hospital, Accra, Ghana, 5 Department of Medicine and Therapeutics, University of Health and Allied Sciences, Ho, Ghana

* taibaafaa@yahoo.com

## Abstract

**Data Availability Statement:** All information required are included in the paper.

**Funding:** The authors received no specific funding for this work.

### Background

Autoimmune hepatitis (AIH) is a progressive inflammatory liver disease of unknown aetiology. The number of reported AIH cases is increasing in the developed countries but the same cannot be said about sub Saharan Africa (SSA). Paediatric AIH diagnosis is usually missed and patients present with decompensated liver disease. Our study highlights the clinical profile of paediatric AIH cases at a referral hospital in Ghana.

### Methods

This is a retrospective review of all cases of children diagnosed with autoimmune hepatitis at the gastroenterology clinic in Korle Bu Teaching Hospital, Accra, Ghana. Data was extracted from the patients' records from April 2016 to October 2019. These children were diagnosed based on the presence of autoantibodies, elevated immunoglobulin G and histologic presence of interphase hepatitis with the exclusion of hepatitis A, B, C and E depending on their clinical presentation, Wilson's disease, HIV, Schistosomiasis and sickle cell disease.

### Results

Thirteen patients aged between 5 years to 13 years with a mean age of 10 years were diagnosed with AIH. All the patients had type 1 AIH with majority 8 (61.5%) being females. Most of the children presented with advanced liver disease with complications. Three patients had other associated autoimmune diseases. The patients were treated with prednisolone with or without azathioprine depending on the severity of the liver disease.

**Competing interests:** The authors have declared that no competing interests exist.

## Conclusion

Majority of paediatric AIH presents with advanced liver disease. There is the need for early detection to change the natural history of AIH in SSA.

## Introduction

Autoimmune hepatitis (AIH) was first reported in the 1950s by Jan Waldenstrom in a group of young women with persistent transaminitis [1]. AIH is a progressive inflammatory liver disease of unknown aetiology that is commoner in females [2]. It is characterized biochemically by elevated transaminases and serum immunoglobulin G (IgG), serologically by non-organic specific and liver specific autoantibodies and histologically by interface hepatitis [3]. The age of onset ranges from infancy to eighth decade of life. However, it is quite uncommon before the age of 2 years, and the incidence is higher between 10 and 30 years [4]. The diagnosis is made after other known causes of liver disease have been excluded and typically there is response to immune suppression [3]. There is associated 40% of family history and 20% of personal history of other autoimmune diseases like thyroiditis, coeliac disease, inflammatory bowel disease, vitiligo, insulin dependent diabetes mellitus and nephrotic syndrome either at presentation or at follow up [5, 6].

There are 2 types of AIH depending on the type of antibodies isolated in serology: antinuclear antibody (ANA) and/or smooth muscle antibody (SMA) in type 1 AIH, and in type 2 there is liver kidney microsomal antibody (anti-LKM) or antibody to liver cytosolic antigen type1 (LC1) [7]. In children, lower autoantibodies levels are considered as positive. For example, ANA and SMA levels of $\geq 1:20$ and anti-LKM of $\geq 1:10$ are positive whilst higher levels of $\geq 1: 40$ are considered positive in adults [4]. Like other autoimmune diseases, AIH is increasing in prevalence over the years. In the United Kingdom, a 6-fold yearly increase was reported over the period between 1990 to the 2000s [8] whilst in Canada, an incidence rate of 0.23 cases per 100,000 children was noted [9]. Not much is known about AIH in African children. To the best of our knowledge only Egypt has two published data in Africa [10, 11] on paediatric AIH. The first study concerned 30 cases from the year 2000 to 2008 [10] and the second, 25 cases [11] over a 13-year period between 2004 and 2017. Both studies had female predominance with type 1 AIH diagnosed in all cases. All patients in both studies had varying degrees of liver disease. Presentations of AIH in children are similar to that seen in adults. Clinical features of AIH vary and can mimic any form of liver disease from fulminant hepatitis to chronic liver disease with features of cirrhosis and portal hypertension [12]. Treatment is predominantly with prednisolone with or without azathioprine [13]. In paediatric AIH, remission is defined as complete clinical recovery with transaminase levels within the normal range, normalization of IgG levels, negative or very low-titre autoantibodies, and histological resolution of inflammation and this depends on the disease severity at presentation [8]. Our study highlights the clinical profile of paediatric AIH cases at a referral hospital in Ghana.

## Materials and methods

### Setting and patients

This is a retrospective review of all cases of children diagnosed with autoimmune hepatitis from April 2016 to October 2019, at the paediatric gastroenterology clinic in Korle Bu Teaching Hospital, Accra, Ghana. The study site is a tertiary hospital that receives referral from all over the country and sometimes from the West African sub region. The diagnosis of AIH was

based on: full history and physical examination, liver chemistry test, coagulation studies (INR), autoantibodies (SMA, ANA, Anti-LKM 1), serum immunoglobulin Ig G levels, serum α1 anti-trypsin levels, abdominal ultrasound and liver biopsy in patient without coagulopathy. The following tests were done to exclude other causes of liver disease: hepatitis B surface antigen, hepatitis C antibody (if antibody tested positive we further confirmed by HCV- RNA), serum α1 antitrypsin levels, Wilson's disease (serum ceruloplasmin, 24 hour urine copper and Keyser-Fleischer rings in the eyes). In patients with acute hepatitis or fulminant hepatitis, hepatitis A and E Ig M was done. HIV 1 and 2, sickle cell disease (Haemoglobin electrophoresis) and Schistosoma Ig M (when there is a history of contact with water bodies). The last 3 investigations above are not recommended in international investigations for AIH, but this is done routinely in Ghana because of the high frequency of these diseases in the population and their possible confounding effect in the diagnosis of AIH. There was no history of intake of any mediation either herbal or orthodox prior to presentation and no family history of autoimmune diseases.

## Data collection

Demographic data, clinical features, biochemical findings, diagnostic imaging, and liver biopsy reports were obtained from the patients' record. Portal hypertension was diagnosed clinically with splenomegaly, ascites and presence of oesophageal varices at endoscopy.

## Statistical analysis

Data collected was entered into excel sheet. These were summarized in terms of frequency and percentage for quantitative data. The median and age range were estimated for the participants in the study.

## Ethical issues

Ethical approval was obtained from the Institutional Review Board (IRB) of the Korle Bu Teaching Hospital. A waiver was obtained from the IRB, as it was difficult to obtain informed consent from the parents or guardians. A 3- letter code was assigned to each patient to ensure anonymity.

## Results

### Demography and clinical features

Thirteen patients with a median age of 10 years (range, 5 to 13 years) and majority females 8 (61.5%) were diagnosed with autoimmune liver disease most of whom were autoimmune hepatitis (Table 1). These patients were being followed up for four and half years to eleven months with a median of two years five months.

### Clinical and laboratory features

Tables 1–3 below show the clinical features, disease outcome and the laboratory investigations done respectively. The duration of symptoms in most but one patient with fulminant hepatitis was between 5 to 24 months with an average of 10 months. The patient with fulminant hepatitis had a duration of symptoms of 1 week and he tested negative for both hepatitis A and E infections. Most patients had chronic liver disease and portal hypertension, with eight patients presenting with hypoalbuminaemia, four with thrombocytopaenia, 8 with anaemia, two with anasarca and three with oesophageal varices, symptomatic with bleeding episodes. The varices were treated with endoscopic variceal band ligation. Histology was done in 9 (69.2%) patients

**Table 1. Clinical features and types of autoimmune diseases.**

| Characteristics (n = 13 patients) | Frequencies |
|---|---|
| **Age at presentation in years** (median [range]) | 10 (5–13) |
| **Duration of symptoms** (median in months) | 5 |
| **Gender, n (%)** | |
| Male | 5 (38.5) |
| Female | 8 (61.5) |
| **Clinical presentation, n (%)** | |
| Fulminant | 1 (7.7) |
| Acute | 2 (15.7) |
| Insidious | 10 (76.6) |
| Complications of chronic liver disease | 8 (61.5) |
| **Detailed clinical findings, n (%)** | |
| Jaundice | 13 (100) |
| Weight loss | 6 (46.2) |
| Bleeding | 3 (23.1) |
| Anasarca | 4 (30.8) |
| Hepatomegaly | 8 (61.5) |
| Splenomegaly | 6 (46.2) |
| Ascites | 5 (38.5) |
| Encephalopathy | 2 (15.4) |
| **Auto-immune disorders, n (%)** | |
| Auto-immune hepatitis (all Type 1) | 11 (84.6) |
| Autoimmune sclerosing cholangitis | 2 (15.4) |
| Systemic lupus erythematosus | 1 (7.7) |
| Ulcerative colitis | 1 (7.7) |

**Table 2. Baseline full blood count, liver function test and INR results at presentation.**

| Patients | Hb (g/dL) | Platelets (x10$^6$/L) | WBC (x10$^6$/L) | INR | Albumin (g/L) | Total Protein (g/L) | ALP (U/L) | ALT (U/L) | AST (U/L) | GGT (IU/L) | Total bilirubin (μmol/L) | Direct bilirubin (μmol/L) |
|---|---|---|---|---|---|---|---|---|---|---|---|---|
| 1 | 10.7 | 151 | 2.9 | 5.1 | 28 | 100 | 577 | 249 | 449 | 118 | 234 | 209 |
| 2 | 9.4 | 228 | 7.2 | 1.7 | 41 | 92 | 639 | 1526 | 1819 | 116 | 315 | 253 |
| 3 | 7.4 | 113 | 3.8 | 8.2 | 33 | 67 | 452 | 279 | 345 | 58 | 284 | 242 |
| 4 | 9.1 | 73 | 4.6 | 5.1 | 19 | 58 | 396 | 155 | 179 | 55 | 251 | 179 |
| 5 | 11.7 | 360 | 7.3 | 1.4 | 41 | 81 | 434 | 1175 | 1090 | 128 | 277 | 138 |
| 6 | 11.6 | 428 | 6.5 | 5.9 | 31 | 55 | 241 | 722 | 1567 | 40 | 551 | 246 |
| 7 | 8.3 | 650 | 4.8 | 4.8 | 21 | 76 | 932 | 473 | 356 | 116 | 376 | 247 |
| 8 | 9.8 | 112 | 3.6 | 3.7 | 28 | 78 | 442 | 531 | 556 | 116 | 219 | 186 |
| 9 | 12.6 | 453 | 5.8 | 1.1 | 43 | 88 | 455 | 864 | 543 | 89 | 244 | 202 |
| 10 | 13.1 | 306 | 5.3 | 4.8 | 34 | 75 | 794 | 3175 | 2025 | 77 | 285.2 | 132 |
| 11 | 9.9 | 180 | 11.4 | 1.4 | 35 | 84 | 381 | 270.9 | 246.9 | 265 | 186.7 | 113 |
| 12 | 9.0 | 342 | 7.1 | 0.9 | 41 | 91 | 873 | 326 | 234 | 96 | 115 | 103 |
| 13 | 9.4 | 137 | 4.2 | 3.9 | 27 | 65 | 227 | 26 | 69 | 75 | 550 | 408 |

ALP: alkaline phosphatase; ALT: alanine aminotransferase; AST: aspartate aminotransferase; GGT: gamma glutamyl transferase; Hb: haemoglobin; INR: international normalised ratio; WBC: white blood cells

**Table 3. Serology and haemoglobin electrophoresis results at presentation.**

| Patients | ANA | SMA | Anti LKM-1 | IgG(6.5–16.0g/L) | HBsAg | Anti-HCV | HCV PCR | Schistosoma IgM | HIV ½ | Hb electrophoresis |
|---|---|---|---|---|---|---|---|---|---|---|
| 1 | Neg | 1:80 | Neg | 26.40 | Neg | Neg | | Pos | Neg | AS |
| 2 | 1:80 | 1:40 | Neg | 22.62 | Neg | Pos | Neg | | Neg | SS |
| 3 | Neg | 1:40 | Neg | 18.64 | Neg | Neg | | | Neg | AA |
| 4 | 1:20 | 1:20 | Neg | 20.60 | Neg | Neg | | Pos | Neg | AA |
| 5 | Neg | 1:20 | Neg | 18.20 | Neg | Neg | | | Neg | AS |
| 6 | Neg | 1:20 | Neg | 20.42 | Neg | Neg | | Pos | Neg | AA |
| 7 | 1:20 | 1:40 | Neg | 17.80 | Neg | Neg | | | Neg | AA |
| 8 | Neg | Neg | Neg | 18. 94 | Neg | Neg | | | Neg | AA |
| 9 | Neg | 1:80 | Neg | 21.10 | Neg | Neg | | | Neg | AS |
| 10 | Neg | 1:80 | Neg | 12.22 | Neg | Neg | | | Neg | AA |
| 11 | Neg | Neg | Neg | 18.80 | Neg | Neg | | | Neg | AA |
| 12 | 1:80 | Neg | Neg | 21.50 | Neg | Neg | | | Neg | AA |
| 13 | Neg | 1:20 | Neg | 12.53 | Neg | Neg | | | Neg | AA |

ANA: antinuclear antibodies; AntiLKM-1: anti liver kidney microsomal type 1; Anti-HCV: hepatitis C antibodies; HCV: hepatitis C virus; HBsAg: hepatitis B surface antigen; Hb: haemoglobin; HIV: human immunodeficiency virus; IgG: immunoglobulin G; IgM: immunoglobulin M; Neg: negative; Pos: positive

whilst the rest had to start treatment without histology due to severe coagulopathy or advanced liver disease (Fig 1).

Four patients had other types of autoimmune disorders. Two patients had autoimmune sclerosing cholangitis or overlap syndrome, after an initial diagnosis of AIH. These patients developed cholestasis during treatment for which Magnetic Resonance Cholangiopancreato-graphy (MRCP) and liver biopsy were done to confirm the diagnosis. One patient had ulcerative colitis, and another had systemic lupus erythematosus making the total of 2 patients with other personal history of non-hepatic autoimmune disorders. There was no family history of autoimmune diseases in these patients. All patients tested negative for Wilson's disease, hepatitis B and HIV. Three patients had specific IgM for *Schistosoma mansoni* and were treated with two courses of praziquantel prior to the diagnosis of AIH. Schistosoma specific Ig M were lost during AIH-specific treatment, as autoantibody levels have normalized.

## Treatment

Prednisolone alone was started in nine patients with severe liver disease or those with portal hypertension at a dose of 2 mg/kg/day (maximum 60 mg/day). This was gradually decreased over a period of three months to a dose of 2.5 mg/day. The liver enzymes together with the INR levels were monitored weekly for two weeks, two weekly for one month, monthly for three months and then three monthly. Azathioprine was added at a dose of 0.5 mg/kg/day, once the liver enzymes had significantly improved. Azathioprine then becomes the maintenance therapy. For patients without severe disease, prednisolone and azathioprine are started at the same time with similar monitoring as in severe cases. None of the patients had stopped treatment at the time of this write up. Ursodeoxycholic acid at a dose of 5mg/kg 3 times daily was added to the treatment for the two patients with primary sclerosing cholangitis.

**Outcome.** All children responded well to initial treatment for AIH. Two patients died during treatment from unrelated causes. The first patient died of acute gastroenteritis four months into treatment after hospital discharge. No oral rehydration therapy was given and this child died on the way to hospital on the third day of the illness. The second patient who died, defaulted hospital follow up for three months, after initial response to prednisolone and he was brought dead to the hospital.

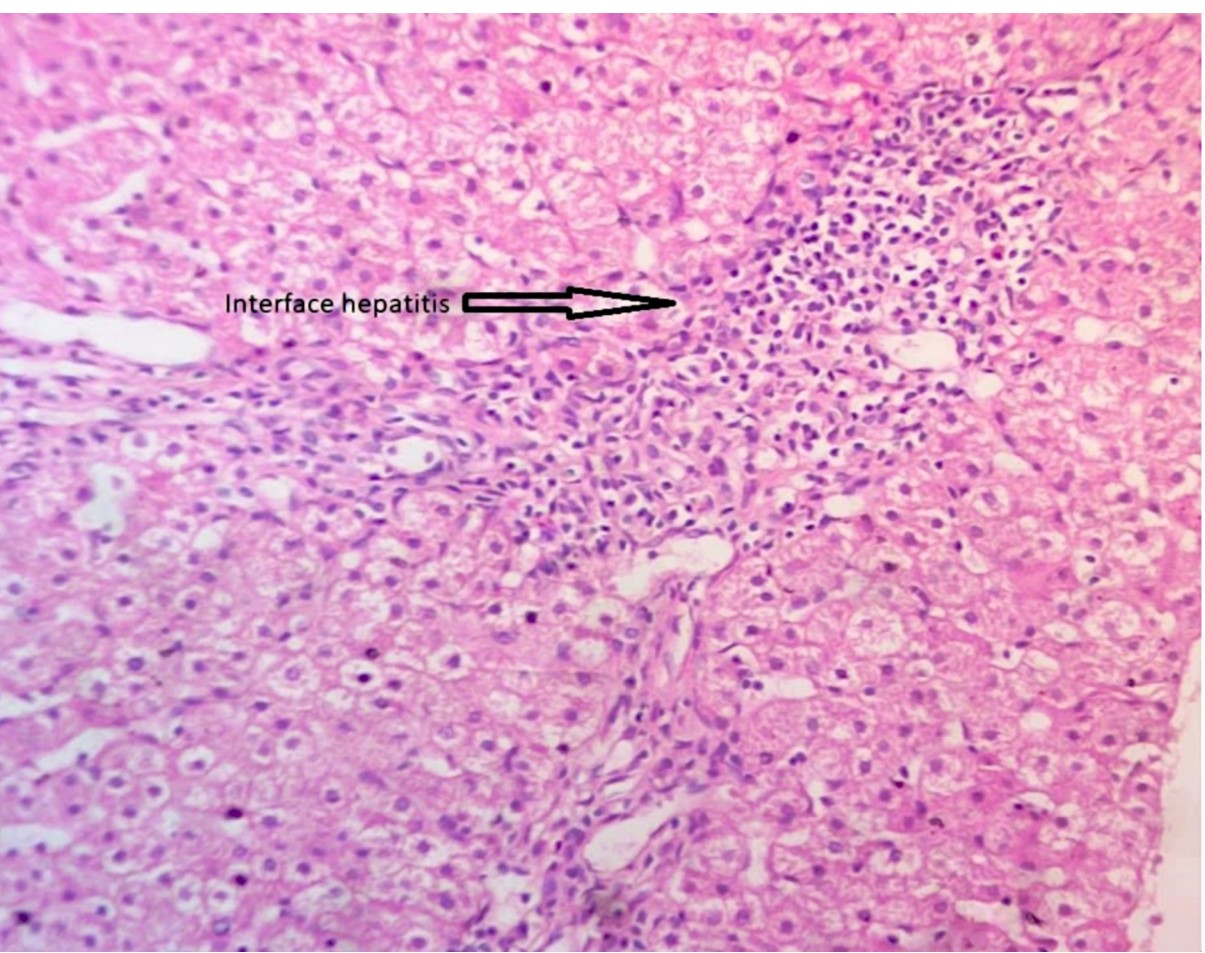

**Fig 1. Histology of autoimmune hepatitis showing interphase hepatitis with periportal inflammation dominated by lymphocytes.**

## Discussion

This is the first report on paediatric AIH from Ghana and possibly from sub Saharan Africa This study, similarly to those from Egypt, did not find any asymptomatic patient as it is usually seen in developed countries [4]. Many odds are against these cohorts with AIH. These are that, autoimmune liver disease in children is inherently aggressive and progresses rapidly with complications [14]. In African Americans, who share a common ancestry with these cohorts, AIH is usually diagnosed at advanced stage with cirrhosis but with good response to therapy [15]. Females predominance of eight (61.5%) patients in this study is similar to that reported in the two Egyptian studies, with 60% [11] and 70.5% [10] respectively and one Canadian study with 60.3% [9]. None of the patients in this study had Type 2 AIH. Type 2 AIH is less common (10%) in paediatric AIH [16] and is usually seen in a small population of younger children [3]. Most of the patients 10 (76.6%) presented insidiously as they had been seen at peripheral health facilities for months without a diagnosis.

We excluded Retroviral infection (HIV) [17], sickle cell disease [18], Schistosomiasis [19] as possible causes of liver disease in our patients because of high prevalence of these conditions in African countries. False positive viral antibodies had been reported in up to 11% of patients with AIH [20]. The positive report of both the hepatitis C virus and Schistosomiasis antibodies in this study may be due to molecular mimicry confounded my hypergammaglobulinaemia in

AIH [21]. The finding of two (15.4%) of these cohorts with normal Ig G level is similar to that reported in international literature where 15% of patients with type 1 AIH had normal Ig G level and are usually with acute presentation [22]. The frequency of other concurrent autoimmune diseases such as ulcerative colitis and systemic lupus erythematosus in this study is similar to the prevalence reported in other studies [23]. The commonest reported extra hepatic autoimmune diseases include, thyroid disease, autoimmune skin diseases, coeliac disease and inflammatory bowel disease [23]. This indicate that all children with autoimmune liver disease should be screened for other autoimmune diseases annually [23].

Liver biopsy and histology required for confirmation and staging of the disease, was done in only nine (69.2%) because of advance liver disease at presentation complicated with coagulopathy, bleeding and ascites, which are contraindications for liver biopsy. In our facility where transjugular liver biopsy is not available, these patients with other laboratory features of AIH were treated with good response similar to what was done in adult studies [24]. All the patients had good response to the standard treatment with prednisolone with or without azathioprine, regardless of the severity of the liver disease [4]. Except for the 2 patients who died, all the remaining patients are still on treatment and are doing well. The circumstances of the mortalities recorded highlight the difficulties in managing patients in developing countries. There is thus, the need for public awareness creation about liver diseases, the importance of seeking healthcare early and remain in care throughout the duration of treatment.

## Limitations of the study

There is small number of patients with short follow up period in this study, as this is not a true reflection of disease prevalence in the country.

## Conclusion

Autoimmune liver disease is also presented in indigenous African children. It should be investigated for in all children with liver disease of any severity, as it is a very treatable condition. Diseases like sickle cell disease and HIV should always be excluded in populations with high prevalence of these diseases. Furthermore, early diagnosis and therapy would reduce liver-related complications.

## Acknowledgments

Dr. Lawrence Edusei of the Department of Pathology of Korle Bu Teaching Hospital for assistance with the histologic reviews and the pictures.

## Author Contributions

**Conceptualization:** Taiba Jibril Afaa, Yaw Asante Awuku.

**Data curation:** Taiba Jibril Afaa, Matilda Tierenye Dono, Eric Odei.

**Formal analysis:** Kokou Hefoume Amegan-Aho, Eric Odei.

**Methodology:** Taiba Jibril Afaa, Matilda Tierenye Dono, Yaw Asante Awuku.

**Resources:** Eric Odei.

**Validation:** Kokou Hefoume Amegan-Aho.

**Writing – original draft:** Taiba Jibril Afaa, Eric Odei.

**Writing – review & editing:** Taiba Jibril Afaa, Kokou Hefoume Amegan-Aho, Yaw Asante Awuku.

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
