## [Decision Letter · Decision Letter 0]

3 Jul 2020

PONE-D-20-09611

Clinical characteristics of paediatric autoimmune hepatitis at a referral hospital in Sub Saharan Africa

PLOS ONE

Dear Dr. Afaa,

Thank you for submitting your manuscript to PLOS ONE. After careful consideration, we feel that it has merit but does not fully meet PLOS ONE’s publication criteria as it currently stands. Therefore, we invite you to submit a revised version of the manuscript that addresses the points raised during the review process.

We look forward to receiving your revised manuscript.

Kind regards,

Nikhil Pai, BSc, MD

Academic Editor

PLOS ONE

Journal Requirements:

2. In the ethics statement in the manuscript and in the online submission form, please provide additional information about the patient records used in your retrospective study.

Specifically, please ensure that you have discussed whether all data were fully anonymized before you accessed them and/or whether the IRB or ethics committee waived the requirement for informed consent.

If patients' guardians provided informed written consent to have data from their medical records used in research, please include this information.

3. Please ensure that you refer to Figure 1 in your text as, if accepted, production will need this reference to link the reader to the figure.  Please also include a caption for figure 1.

4. We note you have included a table to which you do not refer in the text of your manuscript Table 2 is cited but not included. 

5. Table 3 is included but not cited. Please ensure that you refer to Table 3 in your text; if accepted, production will need this reference to link the reader to the Table.

Additional Editor Comments:

Thank you for your submission and your patience with this decision. I would look forward to seeing the comments from the authors thoughtfully addressed and resubmitted. In recognition of the paucity of data from Sub Saharan Africa on this topic, I would encourage the authors to please refine this manuscript further for re-review. Once sent back, we will consider the submission further in light of the requisite edits and issue a further decision.

Reviewers' comments:

Reviewer's Responses to Questions

**Comments to the Author**

1. Is the manuscript technically sound, and do the data support the conclusions?

Reviewer #1: Yes

Reviewer #2: No

2. Has the statistical analysis been performed appropriately and rigorously? 

Reviewer #1: Yes

Reviewer #2: N/A

3. Have the authors made all data underlying the findings in their manuscript fully available?

Reviewer #1: Yes

Reviewer #2: No

4. Is the manuscript presented in an intelligible fashion and written in standard English?

Reviewer #1: Yes

Reviewer #2: No

5. Review Comments to the Author

Reviewer #1: Thank you for writing up your experience with paediatric AIH. There is not much data from Africa and your efforts are applauded. Nonetheless there are some comments that need to be addressed.

Major comments:

- There were a few patients who were ANA negative and ASMA negative with high IgG levels. Indeed IgG levels become elevated in cirrhosis (of any cause). How did the authors confirm a diagnosis of AIH in these children?

- A frequent differential diagnosis for AIH is drug induced liver injury - The authors need to make a comment of whether these patients were taking any medications including herbs/over the counter meds which may have caused DILI

- Could the authors do a Simplified AIH score for their patients? (Hennes et al. Hepatology 2008 Jul;48(1):169-76. doi: 10.1002/hep.22322.).

Can be done at: https://www.mdcalc.com/simplified-autoimmune-hepatitis-aih-score

- The authors mentioned that AIH patients typically present with elevated transaminases. Yet the ALT and AST levels are not presented as well as the other liver function tests (bilirubin, ALP, GGT, albumin) are not presented in the results. Do the authors have this information?

- It is difficult to determine from the results what proportion of patients had a response. This is touched upon in the discussion but really belongs in the results.Typically in Type 1 AIH, 75% of patients are steroid responsive. Also how was the prednisolone weaned over time? What was the maintenance therapy? did patients stay on AZA alone and were weaned off pred? or did they stay on low dose pred? A little more details is needed here

- How long was the median follow-up period for these patients?

- It would be interesting to know if these patients had previously engaged in the hospital (for other reasons) and AIH could have been screened for earlier?

- Did the authors have any information on family history?

- I suggest the authors have subheadings to present the results section rather than 1 whole paragraph.

I.e. Demographics, Clinical and laboratory features, treatment, Outcome

- the section about cause of the 2 deaths belongs in the results section, not the discussion

- The authors should mention briefly about the limitations of the study - e.g. small patient numbers, lack of long follow-up, etc.

Minor comments:

- the acronym SSA for sub Saharan Africa is used without being previously defined in the abstract

- please be consistent with British vs. American spelling. The authors use words like “aetiology” and “paediatric” which is British spelling and then “Normalization” and “titer” which is American spelling.

- the sentence “thirteen patients aged between 5 years to 13 years…” in the Methods should be removed as it belongs in the Results section.

- similarly “Treatment was with steroid with or without azathioprine…” should be removed from the Data collection section”.

- I would change the terms “autoimmune sclerosis cholangitis” and “sclerosing cholangitis” to “primary sclerosing cholangitis”

Reviewer #2: In their retrospective case series “Clinical characteristics of paediatric autoimmune hepatitis at a referral hospital in Sub Saharan Africa» Afaa et al. describe the clinical characteristics of 13 patients.

The introduction

is a general review of AIH, which should be shortened. Instead, I would suggest expanding on clinical features in pediatric AIH described in developed and developing countries (Africa and other parts of the world) in order to lead to a reason why their experience with AIH should be added to the literature.

Results

Did the patient had already cirrhosis? Which stage?

How long was the follow-up time?

Two patient died, please expand on their disease course in the result section. Why did the die?

In table 1

It is stated that 11 patients had AIH and 2 sclerosing cholangitis. Did the later 2 had overlap disease? In the text it is stated that all patients had AIH – please specifiy.

For the expression “hepatic failure” I would suggest using the King’s College criteria for definition of Acute Liver Failure and additionally report the INR value at presentation for all patients.

It would be interesting to document the lab values at presentation. Could the authors provide numbers for liver enzymes (ASAT, ALAT), gGT, albumin, thrombocytes, bilirubin.

For table 2

could the authors state, whether the measured antibodies were negative or not available/ not performed and provide numbers for IgG or at least the multiple of the upper limit of normal.

Please add a legend to the table with the full names of the antibodies, etc.

Discussion

“The positive report of both the hepatitis C virus and Schistosomiasis antibodies are due to the general over production of immunoglobulins in patients with AIH”. – please add a reference for this statement.

In table 2 there are 3 patients with schistosomiasis IgM, in the discussion are only two mentioned, please correct/specify. Why were the patients “treated with 2 courses of praziquantel” when the authors assume that the overproduction of immunoglobulins are responsible for the IgM? – please discuss.

“The finding of 15.4% of these cohorts with normal Ig G level” – as the patients number is low, I would suggest to also state numbers e.g. “Two (15%) patients had normal IgG levels”

Interestingly there are ¾ of patients with insidious onset. In general it is documented to be less than 40% – please discuss.

What is the message of the findings from these 13 patients? The conclusion the authors give is common knowledge.

Some English editing needed.

6. PLOS authors have the option to publish the peer review history of their article (what does this mean?). If published, this will include your full peer review and any attached files.

Reviewer #1: No

Reviewer #2: No

---

## [Author Response · Author response to Decision Letter 0]

10 Sep 2020

Dear Sir,

RESPONSE TO REVIEWERS COMMENTS

Thank you for reviewing my manuscript for possible publication in your journal. Below are the responses to the comments.

Response to the Editor’s Comments

1. Comment: Please ensure that your manuscript meets PLOS ONE's style requirements, including those for file naming. 

Answer: The manuscript naming and style has been changed to PLOS ONE’s style.

2. Comment: In the ethics statement in the manuscript and in the online submission form, please provide additional information about the patient records used in your retrospective study. Specifically, please ensure that you have discussed whether all data were fully anonymized before you accessed them and/or whether the IRB or ethics committee waived the requirement for informed consent. If patients' guardians provided informed written consent to have data from their medical records used in research, please include this information.

Answer: The required information had been added to the manuscript under the section of “ethical Issues”.

3. Comment: Please ensure that you refer to Figure 1 in your text as, if accepted, production will need this reference to link the reader to the figure. Please also include a caption for figure 1.

Answer: Figure 1 had been referred to appropriately in the text and a caption had been added to it.

4. Comment: We note you have included a table to which you do not refer in the text of your manuscript Table 2 is cited but not included. 

Answer: This mistake had been corrected.

5. Comment: Table 3 is included but not cited. Please ensure that you refer to Table 3 in your text; if accepted, production will need this reference to link the reader to the Table. 

Answer: Table 3 has been included and captioned appropriately.

Response to Reviewer 1’s Major Comments

1. Comment: Some patients who were ANA negative and ASMA negative with high IgG levels. IgG levels become elevated in cirrhosis (of any cause). How did the authors confirm a diagnosis of AIH in these children? 

Answer: A combination of laboratory investigations, histology and response to therapy as defined by guidelines were used to diagnose these patients. Several studies have been done in children and adults on sero negative AIH. (Maggiore G, Roux K, Johanet C, Fabre M, Sciveres M, Riva S. Seronegative autoimmune hepatitis in childhood. J Pediatr Gastroenterol Nutr. 2006 ;42(5):E5-E6).

2. Comment: The authors need to make a comment of whether these patients were taking any medications including herbs/over the counter meds which may have caused DILI. These patients were not on any mediation and this comment has been documented in the manuscript. Could the authors do a Simplified AIH score for their patients? (Hennes et al. Hepatology 2008 Jul;48(1):169-76. doi: 10.1002/hep.22322.). Can be done at: https://www.mdcalc.com/simplified-autoimmune-hepatitis-aih-score. Answer: The simplified AIH score was developed for adults. It may not work well for children as very low autoantibody titres (positivity at a dilution ≥ 1:20 for ANA and SMA, ≥ 1:10 for anti-LKM1compared to a dilution ≥ 1:40 in adults) are used for the diagnosis of AIH in children. Moreover, there is no distinction between aih and antoimmune sclerosing cholangitis which can only be differentiated if a cholangiogram is performed. Hence the score will be much lower in these patients. (Ferri PM, Ferreira AR, Miranda DM, Simões E Silva AC.Diagnostic criteria for autoimmune hepatitis in children: A challenge for pediatric hepatologists. World J Gastroenterol. 2012; 18(33): 4470-4473).

3. Comment: The authors mentioned that AIH patients typically present with elevated transaminases. Yet the ALT and AST levels are not presented as well as the other liver function tests (bilirubin, ALP, GGT, albumin) are not presented in the results. Do the authors have this information?

Answer: Yes the information is available and this has been captured in table 2.

4. Comment: It is difficult to determine from the results what proportion of patients had a response. This is touched upon in the discussion but really belongs in the results. Typically in Type 1 AIH, 75% of patients are steroid responsive. Also how was the prednisolone weaned over time? What was the maintenance therapy? did patients stay on AZA alone and were weaned off pred? or did they stay on low dose pred? A little more details is needed here. 

Answer: A detail of the treatment has be documented in the results under treatment and outcome section.

5. Comment: How long was the median follow-up period for these patients? 

Answer: The follow up is from 4 years to 11 months with a median of 2 yrs 5 months.

6. Comment: It would be interesting to know if these patients had previously engaged in the hospital (for other reasons) and AIH could have been screened for earlier? 

Answer: These patients were not previously screened for AIH due to lack of expertise. This is one of the reasons for this publication as data from this part of the world is scanty.

7. Comment: Did the authors have any information on family history? 

Answer: Yes, there was no family history of autoimmune disease in any of the patients.

8. Comment: I suggest the authors have subheadings to present the results section rather than 1 whole paragraph. I.e.Demographics, Clinical and laboratory features, treatment, Outcome. 

Answer: Subheadings have been applied to the results section. 

9. Comment: The section about cause of the 2 deaths belongs in the results section, not the discussion. 

Answer: Causes of death had been moved to results section under outcomes.

10. Comment: The authors should mention briefly about the limitations of the study - e.g. small patient numbers, lack of long follow-up, etc. 

Answer: The limitations of the study has been included. 

Response to the Reviewer 1’s Minor comments

Comment: The acronym SSA for sub Saharan Africa is used without being previously defined in the abstract. 

Answer: The acronym has been applied.

Comment: Please be consistent with British vs. American spelling. The authors use words like “aetiology” and “paediatric” which is British spelling and then “Normalization” and “titer” which is American spelling. 

Answer: Language change has been changed to British spelling. 

Comment: The sentence “thirteen patients aged between 5 years to 13 years…” in the Methods should be removed as it belongs in the Results section. 

Answer: The sentence was removed from the method section.

Comment: Similarly “Treatment was with steroid with or without azathioprine…” should be removed from the Data collection section”. 

Answer: The sentence was removed from the method section.

Comment: I would change the terms “autoimmune sclerosis cholangitis” and “sclerosing cholangitis” to “primary sclerosing cholangitis” 

Answer: An overlap syndrome between AIH and sclerosing cholangitis (ASC) is more common in children than in adults. In paediatric literature ASC or overlap syndrome is the preferred terminology. (Gregorio GV, Portmann B, Karani J, et al. Autoimmune hepatitis/ sclerosing cholangitis overlap syndrome in childhood: a 16-year prospective study. Hepatology 2001;33:544–53).

Response to the Reviewer 2’s Comments

Comment: The introduction is a general review of AIH, which should be shortened. Instead, I would suggest expanding on clinical features in pediatric AIH described in developed and developing countries (Africa and other parts of the world) in order to lead to a reason why their experience with AIH should be added to the literature. 

Answer: Only two studies had been published from Africa and both are from Egypt. Details of these African studies together with other relevant information in paediatrics have been added to the introduction. 

RESULTS

1. Comment: Did the patient had already cirrhosis? Which stage? 

Answer: Four patients had cirrhosis stage 2.

2. Comment: How long was the follow-up time? 

Answer: The follow up is from 4 years to 11 months with a median of 2 yrs 5 months.

3. Comment: Two patient died, please expand on their disease course in the result section. Why did the die? 

Answer: The cause of death and the disease course had been expanded in the result section.

4. Comment: In table 1 It is stated that 11 patients had AIH and 2 sclerosing cholangitis. Did the later 2 had overlap disease? In the text it is stated that all patients had AIH – please specify. 

Answer: Yes the 2 patients had overlap syndrome. All 13 patients were initially diagnosed with AIH 1, until later when the two developed cholestasis, which was confirmed as ASC. 

5. Comment: For the expression “hepatic failure” I would suggest using the King’s College criteria for definition of Acute Liver Failure and additionally report the INR value at presentation for all patients. 

Answer: The King’s College criteria for definition of Acute Liver Failure was used and all the inr levels has been included in table 2.

6. Comment: It would be interesting to document the lab values at presentation. Could the authors provide numbers for liver enzymes (ASAT, ALAT), gGT, albumin, thrombocytes, bilirubin. 

Answer: The lab values at presentation have been documented in table.

7. Comment: For table 2 could the authors state, whether the measured antibodies were negative or not available/ not performed and provide numbers for IgG or at least the multiple of the upper limit of normal. 

Answer: The actual measurements for IgG and autoantibodies levels had been added to table 3.

8. Comment: Please add a legend to the table with the full names of the antibodies, etc. Answer: Legends have been added to the tables

DISCUSSION

1. Comment: The positive report of both the hepatitis C virus and Schistosomiasis antibodies are due to the general over production of immunoglobulins in patients with AIH”. – please add a reference for this statement. 

Answer: The statement has been rephrased and referenced. 

2. Comment: In table 2 there are 3 patients with Schistosomiasis IgM, in the discussion are only two mentioned, please correct/specify. 

Answer: Three patients tested positive for Schistosomiasis as stated in the result, and this has been corrected in the discussion.

3.Comment: Why were the patients “treated with 2 courses of praziquantel” when the authors assume that the overproduction of immunoglobulins are responsible for the IgM? – please discuss. 

Answer: The patients were treated with two courses of praziquantel prior to the involvement of the author in the management of the patients. The issue of overproduction of immunoglobulins as been discussed.

4. Comment: “The finding of 15.4% of these cohorts with normal Ig G level” – as the patients number is low, I would suggest to also state numbers e.g. “Two (15%) patients had normal IgG levels” 

Answer: This has been corrected. 

5. Comment: Interestingly there are ¾ of patients with insidious onset. In general it is documented to be less than 40% – please discuss. 

Answer: This has been discussed.

6. Comment: What is the message of the findings from these 13 patients? The conclusion the authors give is common knowledge. 

Answer: The conclusion has been reviewed.

7. Comment: Some English editing needed. 

Answer: The language has been edited.

Thank you

Taiba Jibril Afaa.

---

## [Editor Report · Decision Letter 1]

17 Sep 2020

Clinical characteristics of paediatric autoimmune hepatitis at a referral hospital in Sub Saharan Africa

PONE-D-20-09611R1

Dear Dr. Afaa,

We’re pleased to inform you that your manuscript has been judged scientifically suitable for publication and will be formally accepted for publication once it meets all outstanding technical requirements.

Kind regards,

Nikhil Pai, BSc, MD

Academic Editor

PLOS ONE

Additional Editor Comments (optional):

Thank you for your responses to our reviewers.
---

## [Editor Report · Acceptance letter]

22 Sep 2020

PONE-D-20-09611R1 

Clinical characteristics of paediatric autoimmune hepatitis at a referral hospital in Sub Saharan Africa 

Dear Dr. Afaa:

I'm pleased to inform you that your manuscript has been deemed suitable for publication in PLOS ONE. Congratulations! Your manuscript is now with our production department. 

Kind regards, 

on behalf of

Dr. Nikhil Pai 

Academic Editor

PLOS ONE